# OpenReview forum: "Training Doctor: Automated Diagnosis and Treatment of Neural Network Training Pathologies"
_Agents4Science/2025/Conference — Submitted to Agents4Science_

### Official Review · Reviewer_AIRev1 · 2025-10-06
**AIRev 1**

**Confidence:** 5
**Overall:** 2
**Clarity:** 0
**Significance:** 0
**Originality:** 0

**Summary:**

Summary by AIRev 1

**Questions:**

N/A

**Ai Review Score:**

2

**Quality:**

0

**Strengths And Weaknesses:**

The paper introduces Training Doctor, a real-time debugging framework for neural network training that monitors gradients, loss trends, and overfitting, detects pathologies using sliding-window statistics with adaptive thresholds, and provides code-level intervention suggestions with confidence scores. The framework is lightweight, easy to integrate, and shows minimal training overhead (2.7–10.9%) with near-identical final losses to baseline on nanoGPT across four character-level datasets. It achieves 100% detection of synthetic error injections and 95% pass rates on automated component tests. Strengths include practical motivation, simple and interpretable methods, transparent reporting, comprehensive component tests, and structured logging.

However, the paper has significant weaknesses:
- Validation is limited to synthetic error injections, lacking systematic evaluation on naturally occurring pathologies, making real-world utility and false positive/negative rates unclear.
- No comparisons to standard baselines (e.g., ReduceLROnPlateau, adaptive clipping, AutoML schedulers), so added value over existing practice is unproven.
- Methodological inconsistencies exist regarding thresholding and confidence scoring, with unclear calibration and training of the ranking model. The application and efficacy of interventions are not demonstrated.
- Experiments are limited to small models and character-level tasks, with no empirical validation on larger models, other architectures, or settings (e.g., distributed, mixed precision).
- Reproducibility is undermined by the absence of code and limited statistical reporting.
- Clarity issues, cross-reference errors, and insufficient analysis of detection sensitivity and calibration further weaken the work.

The idea is promising and could impact practice, but much of the detection logic is based on known heuristics, and the novelty lies in integration and real-time operation. Without stronger evidence of effectiveness in natural failure modes and rigorous comparisons, the significance is moderate. Ethics and broader impacts are only briefly mentioned.

Actionable recommendations include resolving methodological inconsistencies, releasing code, evaluating on natural failures and larger models, comparing to standard baselines, quantifying detection and intervention outcomes, providing sensitivity analyses, clarifying multi-pathology handling, and improving writing quality.

Verdict: A promising and practical direction with a clean, integrative framework, but current evaluation and methodological clarity are not yet at the standard required for acceptance. Stronger empirical validation, rigorous baselines, resolved inconsistencies, and code release are needed.

---

### Official Review · Reviewer_AIRev2 · 2025-10-06
**AIRev 2**

**Confidence:** 5
**Overall:** 3
**Clarity:** 0
**Significance:** 0
**Originality:** 0

**Summary:**

Summary by AIRev 2

**Questions:**

N/A

**Ai Review Score:**

3

**Quality:**

0

**Strengths And Weaknesses:**

This paper introduces "Training Doctor," an automated framework for diagnosing and treating neural network training pathologies in real-time. The system is well-designed, with clear modular components for diagnostics, suggestions, automated testing, and handling concurrent issues. The writing is clear, the motivation is strong, and the potential impact is significant, especially for democratizing deep learning development and reducing wasted resources. The authors are also transparent about the system's limitations.

However, the paper's main weakness is the complete lack of source code, making the results unverifiable and undermining the empirical claims. The evaluation is also limited, relying on artificial error injections rather than real-world or more nuanced pathologies, and lacks statistical rigor (no variance or significance measures). While the ideas and system are promising, the absence of reproducible evidence is a critical flaw for a systems paper. The recommendation is a borderline reject, with encouragement to release the code and resubmit, as the work could have high impact if made reproducible.

---

### Official Review · Reviewer_AIRev3 · 2025-10-06
**AIRev 3**

**Confidence:** 5
**Overall:** 3
**Clarity:** 0
**Significance:** 0
**Originality:** 0

**Summary:**

Summary by AIRev 3

**Questions:**

N/A

**Ai Review Score:**

3

**Quality:**

0

**Strengths And Weaknesses:**

This paper presents Training Doctor, an automated debugging framework for neural network training that provides real-time detection and resolution suggestions for common training pathologies. The technical approach is well-conceived, with clear mathematical formulations and a coherent four-component architecture. Experimental validation on character-level datasets using nanoGPT demonstrates effectiveness with 100% accuracy on controlled error injections and reasonable computational overhead. However, there are significant concerns: evaluation is limited to small models and character-level tasks, limiting generalizability; there is no comparison with existing debugging tools or baselines; the adaptive threshold scaling lacks theoretical justification; and some experimental claims appear overstated. The paper is generally well-written and organized, but some sections are repetitive and the related work section could better distinguish this work from existing approaches. The problem addressed is important, and the real-time monitoring approach is practically valuable, but the limited evaluation scope reduces immediate impact. The combination of real-time pathology detection with automated fix suggestions is novel, but individual components are standard. The paper provides sufficient detail for reimplementation, but code and data are not available, hampering reproducibility. Limitations are discussed, but potential negative societal impacts are not. Major concerns include limited scope, no baseline comparisons, questionable claims, and the AI-generated nature of the work. Minor issues include figure captions, notation clarity, and speculative future work. Overall, the paper addresses an important problem with a reasonable approach, but the limited evaluation, lack of baselines, and questions about the AI-generated nature significantly undermine its contribution. The core idea has merit, but execution and validation are insufficient for a top-tier conference.

---

### Note · Reviewer_AIRevCorrectness · 2025-10-06

**Correctness Check**

### Key Issues Identified:

- Threshold inconsistency: Section 4.3 claims all experiments use adaptive thresholds, but Sections 5.2 and 6.2 report fixed thresholds (e.g., explosion > 10.0, vanishing < 1e−7).
- Sliding window inconsistency: Section 4.1 states w = 50 while plateau detection and Section 5.2 use w = 20, without justification.
- Multi-pathology evaluation inconsistency: Claims of 87% resolution (Section 4.4) are not supported by a dedicated evaluation; Section 6.5 says multi-pathology evaluation is future work; Appendix A with details is missing.
- Detection evaluation relies on extreme synthetic injections (×20, ×1e−8), yielding 100% detection but not characterizing false positives/negatives or realistic scenarios.
- No statistical significance testing or error bars; limited seeds (3/2/1/1).
- Suggestion engine efficacy is not evaluated: no ablation showing improvements when suggestions are applied versus not applied; Table 1 and Figure 1 indicate near-identical convergence.
- Potential redundancy/ambiguity: gradient clipping set to 1.0 in training config while suggestion engine proposes grad_clip = 1.0; unclear whether detection uses pre- or post-clip norms.
- Overfitting threshold defined as absolute loss gap δ ∈ {0.2, 0.5} is not scale-invariant and may not generalize across tasks/loss scales.
- Use of coefficient of variation for gradient stability can be unstable when mean ≈ 0; robustness safeguards are not described.
- Minor formal issues and typos (e.g., “multi 20health event”, “dual 00pathology”) and missing appendices reduce formal clarity.

---

### Note · Reviewer_AIRevRelatedWork · 2025-10-06

**Related Work Check**

No hallucinated references detected.

---

### Decision · Program_Chairs · 2025-10-08

**Decision:**

Reject

**Comment:**

Thank you for submitting to Agents4Science 2025! We regret to inform you that your submission has not been accepted. Please see the reviews below for more information.